# Diet/Nutrition: Ready to Transition from a Cancer Recurrence/Prevention Strategy to a Chronic Pain Management Modality for Cancer Survivors?

**DOI:** 10.3390/jcm11030653

**Published:** 2022-01-27

**Authors:** Sevilay Tümkaya Yılmaz, Anneleen Malfliet, Ömer Elma, Tom Deliens, Jo Nijs, Peter Clarys, An De Groef, Iris Coppieters

**Affiliations:** 1Pain in Motion Research Group (PAIN), Department of Physiotherapy, Human Physiology and Anatomy, Faculty of Physical Education and Physiotherapy, Vrije Universiteit Brussel, 1090 Brussels, Belgium; sevilay.tumkaya.yilmaz@vub.be (S.T.Y.); Anneleen.Malfliet@vub.be (A.M.); omer.elma@vub.ac.be (Ö.E.); Jo.Nijs@vub.be (J.N.); 2Pain in Motion International Research Group, 1090 Brussels, Belgium; an.degroef@kuleuven.be; 3Research Foundation Flanders (FWO), 1000 Brussels, Belgium; 4Department of Physical Medicine and Physiotherapy, University Hospital Brussels, 1090 Brussels, Belgium; 5Department of Movement and Sport Sciences, Faculty of Physical Education and Physiotherapy, Vrije Universiteit Brussel, 1050 Brussels, Belgium; Tom.Deliens@vub.be (T.D.); Peter.Clarys@vub.be (P.C.); 6Institute of Neuroscience and Physiology, Unit of Physiotherapy, Department of Health & Rehabilitation, University of Gothenburg, 40530 Gothenburg, Sweden; 7Department of Rehabilitation Sciences, Research Group for Rehabilitation in Internal Disorders, KU Leuven, 3000 Leuven, Belgium; 8Department of Rehabilitation Sciences, MOVANT Research Group, University of Antwerp, 2000 Antwerp, Belgium; 9Laboratory for Brain-Gut Axis Studies (LaBGAS), Translational Research Center for Gastrointestinal Disorders (TARGID), Department of Chronic Diseases, Metabolism, and Ageing, KU Leuven, 3000 Leuven, Belgium

**Keywords:** cancer survivors, chronic pain, pain management, nutrition, diet

## Abstract

Evidence for the relationship between chronic pain and nutrition is mounting, and chronic pain following cancer is gaining recognition as a significant area for improving health care in the cancer survivorship population. This review explains why nutrition should be considered to be an important component in chronic pain management in cancer survivors by exploring relevant evidence from the literature and how to translate this knowledge into clinical practice. This review was built on relevant evidence from both human and pre-clinical studies identified in PubMed, Web of Science and Embase databases. Given the relationship between chronic pain, inflammation, and metabolism found in the literature, it is advised to look for a strategic dietary intervention in cancer survivors. Dietary interventions may result in weight loss, a healthy body weight, good diet quality, systemic inflammation, and immune system regulations, and a healthy gut microbiota environment, all of which may alter the pain-related pathways and mechanisms. In addition to being a cancer recurrence or prevention strategy, nutrition may become a chronic pain management modality for cancer survivors. Although additional research is needed before implementing nutrition as an evidence-based management modality for chronic pain in cancer survivors, it is already critical to counsel and inform this patient population about the importance of a healthy diet based on the data available so far.

## 1. Introduction

The Survivorship Task Force describes cancer survivors as “all people who have been diagnosed with cancer, who have finalized primary cancer treatment (except the maintenance therapy, like immune and hormone therapy) and have no mark of active disease” [1]. With earlier diagnosis and improvements in treatment, cancer patients are more likely to survive the disease and therefore live longer [1,2], which is reflected in an increased prevalence of cancer survivors over the last 40 years [1]. Yet, although cancer survivors are considered disease-free, they often suffer from physical, social, and emotional problems that severely influence their quality of life [2].

In the cancer survivor population, the development of chronic pain (“pain that continues beyond the expected healing time” [3]) is one of the most often seen sequelae [4]. Pain is reported in 39.3% of survivors after curative treatment [5]. Severe chronic pain associated with a decrease in function is seen in 5 to 10% of survivors [6]. Moreover, pre-existing pain, repeated surgery, psychological vulnerability, radiation therapy, chemotherapy, sociodemographic and psychosocial (depression, anxiety, sleep disturbance, etc.) profiles, hormone therapy, body mass index (BMI) > 30 kg/m^2^ are some of the “predisposing factors” for chronic pain in cancer survivors [4,7,8,9].

Besides chronic pain, cancer survivors often show significant nutritional deficiencies which crucially impact their quality of life [10]. Changed taste, anorexia, unintended weight loss, and, in certain cases, increased adiposity or obesity are all nutrition-related complications that may develop as a result of cancer and its treatment (i.e., chemo-radiotherapy etc.) since the systemic nature of cancer promotes metabolic dysregulation, increased catabolism, and even cachexia [11]. Significantly, an increase in body weight (or obesity) often occurs during cancer treatment and is related to a higher chance for comorbidities (including chronic pain) [12,13]. For that reason, besides a solution for pain, cancer survivors often look for information on nutrition and diet supplements to improve treatment outcome, quality of life and to increase their long-term survival rates [14].

Interest in the link between chronic pain and nutrition has increased tremendously in recent years. On the one hand, recent research indeed revealed that nutritional aspects can influence brain plasticity and function, and therefore may influence central nervous system health and disease (i.e., central sensitization) [15]. On the other hand, as shown by local and widespread pressure hypersensitivity and hyperalgesia, central sensitization occurs in survivors of breast [16], colon [17] and head and neck cancer [18].

Persistent pain in cancer survivors is often complex (neuropathic, nociplastic, and/or nociceptive) in nature [19], underrecognized, undertreated and less responsive to regular chronic pain management approaches (i.e., pharmacological treatments, rehabilitation, etc.) [7]. Additionally, in long-term survivors (in comparison to people without a history of cancer), it is known that the incidence and relative risk of chronic comorbidities is high [20], which result in significantly more functional limitations and pain intensity, making them less likely to respond to standard chronic pain treatment [21]. Despite crucial medical advances, multi-modal pain management approaches, and enhanced survival, the majority of cancer survivors with pain stated that their pain was only alleviated by 61% [22]. Patients and health care providers are frequently not aware of other possible rehabilitation approaches (like pain education, mind-body interventions) and their potential benefits in the pain management during and after cancer treatment [23]. Recently, the ability of daily diet to modulate pain onset and peripheral analgesic sensitivity has come to light as physicians are more attentive to the lifestyle of patients to better prevent side effects such as chronic pain onset, the main reason of medical intervention, healthcare costs, and outpatient counseling [24]. Nutritional medicine for treating persistent pain requires a comprehension of the disease process’ pathogenesis that helps practitioners to prescribe ingredients with particular roles in alleviating the disease process, such as inflammation reduction, or with particular influences on other factors which contribute to pain (i.e., stress and insomnia) [25], or oxidative stress-modulating compounds and oxidative stress status [26].

Despite the increased awareness of the high prevalence, little research has been performed on chronic pain in the cancer population, leading to an important knowledge gap and a lack of clear management guidelines [22]. The most recent systematic review of the association between chronic pain and nutrition in cancer patients and survivors found no evidence in cancer survivors [27], which clearly identifies the knowledge gap and need to address this issue conceptually and scientifically. For all these reasons, this narrative review discusses the mechanisms involved in chronic pain and the critical interaction between these systems and survivors’ diets. In addition, dietary interventions that might provide sustainable, long-term, self-manageable and cost-effective implications for chronic pain management in cancer survivors are proposed. It is shown that nutrition holds potential to become a chronic pain management modality for cancer survivors.

## 2. Methods

This narrative review was accomplished by looking for both pre-clinical and human studies in PubMed, Web of Science and Embase databases by combining the terms “chronic pain”, “cancer survivors” and “nutrition” or “diet”. English-language articles were accessed until September 2021. The reference lists from the articles that were retrieved were also carefully searched.

## 3. Pain and Nutrition in Cancer Survivors: An Update from Cancer and Chronic Pain Literature

Since both nutritional and chronic pain mechanisms and pathways are known for their complexity, it does not come as a surprise that research covering the link between both is complex, ambiguous and involves different explanatory components [24]. Dietary factors (like food preparation, food processing and dietary patterns) exert their impact by several pathways and mechanisms such as glucose-insulin homeostasis, blood lipids, blood pressure, functions of the endothelial, cardiac and adipocyte systems, the gut microbiome, systemic inflammation, and hunger and satiety [28]. Chronic pain and its treatment (like opioids), in turn, are known to have an interplay with the nervous systems and the immune system [29,30]. This narrative piece shines a light on the following different pathways and mechanisms to link diet/nutrition and (chronic) pain in cancer survivors (Figure 1): (1) through obesity; (2) through malnutrition, nutritional deficiency, and diet quality; (3) through the immune system and systemic inflammation; and (4) through gut microbiota.

### 3.1. Impact of Diet and Nutrition on Pain in Cancer Survivors through Obesity

Obesity continues to be a major public health concern, and it is especially frequent among cancer patients, thus determining its long-term impact on the expanding population of cancer survivors is critical [31]. Moreover, it has been known since the 1970s that women with breast cancer receiving adjuvant chemotherapy experience weight gain, commonly reported as 2–5 kg but with great variance [32]. Obesity, defined as a BMI of 30 kg/m^2^ or higher, is also a common risk factor for poor health-related quality of life in cancer survivors, in particular colorectal, breast, and prostate cancer survivors [33].

Obesity can cause chronic pain through two primary processes: mechanical stress, which occurs when extra body weight puts stress on joints in the musculoskeletal system, and systemic proinflammatory state, which is linked to adipose tissue and can increase pain [34]. Obesity (particularly caused by excessive abdominal fat) is related to an increment in chronic systemic inflammation which can have a contribution to central sensitization [35]. According to Emery et al., a diet rich in anti-inflammatory foods, such as the consumption of seafood and plant protein, appears to be linked to the relationship between body fat and pain ratings in healthy adults so they advocated that diet can be addressed as part of pain treatment and evaluation, specifically among overweight and obese people [36]. Additionally, obesity has been linked to microbial homeostasis distortion, with a decrease in bacterial biodiversity and altered expression of bacterial genes, particularly those involved in dietary energy extraction [37]. The increased understanding of the interactions between the gut microbiota and the central nervous system, also known as the gut-brain axis, makes the hypothesis of the gut microbiota’s possible effect on the pain processing and the pain perception reasonable [38], so does obesity. With good-to-moderate patient-centered evidence, the most recent review (*n* = 26) in taxane and platinum-treated cancer patients found a link between obesity and increased severity or occurrence of chemotherapy-induced peripheral neuropathy (CIPN) [39]. Additionally, weight gain (>5%) following breast cancer was found to be positively associated with above-average pain [40].

However, evidence on the relationship between obesity and (chronic) pain in cancer survivors remains very limited. One study found a correlation between a higher BMI and a lower physical quality of life in cancer survivors, including more pain even after taking into consideration age, race, education level, cancer type, and comorbidities [41]. Again, in a meta-analysis conducted by Leysen et al. [4], among other factors, a BMI of 30 kg/m^2^ or higher was substantially linked with the development of chronic pain in breast cancer survivors. In another study, it has been suggested that cancer survivors with CIPN and co-occurring obesity may be more at risk of lower quality of life due to higher symptom severity and pain than non-obese survivors [33]. In parallel, among cancer survivors with CIPN who received platinum and/or taxane chemotherapeutic compounds, overweight and obese survivors experienced more severe pain and higher pain interference scores than normal-weight survivors [42]. Similar to that overweight or obese breast cancer survivors with weight loss of ≥ 5% showed improvement in their pain at 12 months, but these changes were not significantly different from those who lost < 5% [43]. Therefore, weight reduction techniques for obese cancer survivors suffering from chronic pain could be a key factor within pain management for this population. Moreover that studies examining whether dietary management results in pain relief in cancer patients receiving chemotherapy or in survivors after treatment (or in any other cancer treatment associated with pain) are urgently needed and represent an important research priority.

### 3.2. Impact of Diet and Nutrition on Pain in Cancer Survivors through Malnutrition, Nutritional Deficiency, and Diet Quality

Cancer patients suffer from a large catabolic imbalance which causes weight loss, the key indicator of cancer-associated malnutrition [44]. The prevalence of malnutrition is estimated to be between 50 and 80%, depending on the tools used and the populations studied [45], and can reach up to 85% of patients with certain cancers such as pancreatic [46]. According to many proposed mechanisms, which varying from signaling molecules included within the diet (such as oxidized lipids), vagus nerve activation, microbiota alterations, and oxidative stress to maladaptive neuroplasticity induced by hyper-palatable energy-dense foods, poor nutrition may also cause activation of the immune system, in particular by glial activation with increased inflammation and nervous system hypersensitivity as a consequence [47]. Available data clearly revealed that well-nourished breast cancer survivors had improved functions and less symptoms including pain in comparison to malnourished breast cancer survivors [48].

Additionally, nutritional reduction is frequently acknowledged as a component of the cancer course and treatment [49]. During chemotherapy, compared to women without cancer, breast cancer patients reported a significantly lower absolute protein, fat, and alcohol intake, but not carbohydrates and fiber. [50]. In Iranian breast cancer survivors, the average daily energy intake was lower than the estimated energy requirement as a reference value, with just 34% of participants meeting the estimated energy requirement, whereas the mean intakes of vitamin D, vitamin E, iron, and magnesium were insufficient to meet the Food and Nutrition Board’s (1997–2001) guideline of dietary reference intakes [48].

In cancer patients, both at the time of diagnosis and during treatment, micronutrient deficits are common [11]. For instance, pancreatic cancer patients who have had their pancreas removed are at risk of many nutrition-related comorbidities, including an impact on gastrointestinal and hepatic function, glycaemic regulation, bone health, and the status of many micronutrients such as vitamin A, B12, D, E, iron, magnesium and zinc [51]. Similarly, one possible metabolic consequence after a gastrectomy after gastric cancer is vitamin B12 deficiency, which may lower cancer survivors’ quality of life [52].

Importantly, low macro/micronutrient consumption, particularly omega-3 fatty acids, vitamins B1, B3, B6, B12, and D, magnesium, zinc, and -carotene, is associated with chronic neuropathic or inflammatory pain [53]. As seen in multiple systematic studies on various pain conditions, including aromatase inhibitor(AI)-related arthralgia in breast cancer [54], supplementing the diet with these specific nutrients helps to alleviate chronic pain [55]. For example, since estrogen increases vitamin D receptor activation, a low estrogen status could potentially reduce the available active vitamin D amount; 75 to 90% of women receiving AI therapy have a vitamin D deficiency [56], which might negatively contribute to a chronic pain state [57]. It is known that vitamin D deficiency causes a muscle and joint aches syndrome similar to Aromatase Inhibitor-Induced Arthralgias (AIA) [56]. As a result, it is suggested that vitamin D can have a crucial role in several cellular activities considered preventive against the development and modulation of chronic pain [57].

Interestingly, cancer and its treatment may increase the requirement for antioxidant nutrient intake such as vitamin C because of the increased free radicals [11]. Administrating some anti-cancer therapies has shown a significant reduction in patients’ vitamin C concentrations and report of scurvy (vitamin C deficiency disease)-like symptoms so cancer patients are one of the many patient groups who have a high prevalence of hypovitaminosis C and vitamin C deficiency [58]. The mini-review that reviewed the few current trials exploring the impact of IV vitamin C on cancer- and chemotherapy-related quality of life discovered considerable reductions in pain following vitamin C administration [59].

Additionally, it is known that magnesium supplementation is used in a variety of neuropathic pain situations, including cancer-related neuropathic pain and chemotherapy-related neuropathy, as is shown in a recent review that looked for nutritional supplements for the treatment of neuropathic pain [60]. Still, evidence supporting magnesium supplementation for the treatment of (neuropathic pain) following cancer is lacking.

Similarly, short-chain fatty acids (SCFAs) are essential mediators of pain since they fundamentally modulate inflammation [24]. In a network meta-analysis with randomized controlled trials included in the six systematic reviews, Kim et al. [54] showed that omega-3 fatty acids are one of the treatment modalities which attained significant improvement in pain severity compared to wait list controls in breast cancer survivors with AIA. However, since the overall confidence level of each review was limited, no recommendations can be made at present to reduce pain in patients with AIA [54].

Apart from this, several studies have found a link between cancer survivors’ health-related quality of life and their adherence to general non-cancer-specific dietary guidelines, such as the Healthy Eating Index and the Mediterranean diet [41,61]. Higher adherence to the traditional Mediterranean Diet (high consumption of plant-based foods (vegetables, fruit, whole grains, legumes, nuts, olive oil) and low or limited consumption of red meat, milk, and sweets) were linked to higher physical functioning and health status, as well as lower pain and insomnia symptoms, suggesting that this diet may play a role in the quality of life of recently diagnosed female breast cancer patients [62]. Likewise, Wayne et al. [63] claimed that women newly diagnosed with first primary breast cancer (in situ or stage I to IIIA disease) with excellent diet quality according to the Diet Quality Index received higher quality of life scores than women with poor diet quality, including physical health subscale category “bodily pain” with the highest scores.

Furthermore, cancer survivors are advised to follow some diet recommendations from the American Cancer Society (ACS) Guideline on Diet and Physical Activity for Cancer Prevention and the World Cancer Research Fund (WCRF)/American Institute for Cancer Research (AICR) Cancer Prevention (Table 1). Higher diet scores were associated with many aspects, including bodily pain among breast cancer survivors with stage II–III cancer, according to a cross-sectional study that looked at whether adherence to the American Cancer Society (ACS) guidelines was associated with health-related quality of life (HRQoL) among Korean breast cancer survivors [64]. A study of Chinese patients with breast cancer who followed the WCRF/AICR guidelines (BMI, physical activity, and diet) before and after their diagnosis found that following the BMI prescription resulted in reduced pain scores while adherence to dietary recommendations, on the other hand, was not linked to pain scores [65].

### 3.3. Impact of Diet and Nutrition on Pain in Cancer Survivors through the Immune System and Systemic Inflammation

Chronic pain frequently arises from a permanent pro-inflammatory state [55]. The proposed pathophysiology and mechanisms that maintain chronic pain emerge constantly, yet as part of the maladaptive synaptic plasticity related to chronic pain, proposing permanent low-grade inflammation (neuroinflammation) as a primary driver makes therapeutic approaches targeting immune activation to reduce the pro-inflammatory state important to consider for chronic pain management [47].

This pro-inflammatory state is also a characteristic of cancer. Independent of the increment in neural density noticed in the tumor environment, numerous pain modulating agents such as hydrogen ions, tumor necrosis factor-alpha (TNF-α), transforming growth factor-beta (TGF-β), prostaglandins, interleukin-1 (IL-1) and IL-6 are set free into the tumor vicinity, sensitizing and stimulating sensory fibers, possibly contributing to neuronal hyperexcitability and pain [1].

Dietary components have the potential to have substantial inflammatory or anti-inflammatory features [68]. Inflammation is linked to dietary consumption of omega-3 and omega-6 polyunsaturated fatty acids (PUFAs) in healthy populations, according to observational studies since higher omega-3 PUFAs are associated with lower levels of pro-inflammatory indicators such as interleukin (IL)-6, IL-1 receptor antagonist, TNF-α, and C-reactive protein (CRP), as well as higher levels of anti-inflammatory indicators such as IL-10 and transforming growth factor β [69]. Additionally, it is known that within the neurological system, certain combinations of omega-3 and micronutrients (like vitamin A and D) may show an even bigger synergistic effect on inhibiting microglial-mediated neuroinflammation [70]. Similarly, high consumption of dietary fibers is inversely related to the circulating inflammatory markers interleukin 6 (IL-6) and tumor necrosis factor α receptor 2 (TNF-α-R2) in postmenopausal women and C-reactive protein (CRP) in breast cancer survivors [71]. Moreover, a diet high in fruit, vegetables, whole grains, white meat, tomato, legumes, tea, and fruit juices is substantially and inversely related to indicators of systemic inflammation whereas consumption of refined cereals, red meat, butter, processed meat, high-fat dairy, sweets, desserts, pizza, potatoes, eggs, hydrogenated fats, and soft drinks, are found to be strongly and positively associated with systemic inflammation [72].

Looking at this matter from a dietary level rather than a nutritional level, the Mediterranean diet has a high anti-inflammatory micronutrients and phytochemical content such as n-3 fatty acids, flavonoids, carotenoids, and vitamins C and E [73]. Evidence shows that higher adherence to the Mediterranean diet is linked to a lower inflammatory status [74]. As a result, applying an intervention to increase the adherence to a Mediterranean diet pattern may have health benefits by reducing systemic inflammation [73]. In this regard, also more general, several studies have linked diet quality to inflammation. For example, breast cancer survivors with better postdiagnosis diet quality showed lower CRP levels (1.6 mg/L vs. 2.5 mg/L) and higher scores on the Healthy Eating Index (2005) [75]. Likewise, Orchard et al. observed that a higher HEI-2010 score was strongly associated with reduced IL-6 and TNFR-2 levels in breast cancer survivors [76].

Another approach to affect the nervous system’s neuroimmune function is through metabolic alterations [47]. According to mounting evidence, oxidative stress can activate and maintain pain pathways via activating glutamatergic transmission and numerous inflammatory pathways (which are important for the development of peripheral and central sensitization), as well as directly influencing nociceptive centers in the brain [77]. Oxidative stress has been proven to be a significant contributor to the pain caused by chemotherapy-induced peripheral neuropathy (CIPN) [78]. The findings in mouse models showed that cisplatin-induced mechanical hypersensitivity is caused by peripheral oxidative stress sensitizing mechanical nociceptors, whereas paclitaxel-induced mechanical hypersensitivity is caused by central (spinal) oxidative stress maintaining central sensitization that abnormally produces pain in response to Aβ fiber inputs [78]. Furthermore, mitochondrial dysfunction caused by cancer cells (induced by the mitochondrial genome alterations, the associated oxidative stress etc.) [79] may play a role in chronic pain. Maintaining mitochondrial function has been proposed as a possible treatment technique for treating or preventing chronic pain [80]. For example, strategies that improve mitochondrial function have shown success in preventing and reversing CIPN in pre-clinical animal models and have begun to show some progress toward translation to the clinic [81]. Although dietary intake ultimately directs metabolism, only a few studies showed how metabolic pathways influenced by diet may have a role in the immune activation seen in chronic pain [82]. It is asserted that diets high in fruit and vegetable consumption can decrease oxidative stress [72]. For example, antioxidants produced from food, such as vitamin A, CoQ10, vitamin E, and vitamin C, have been demonstrated to play an important role in preventing oxidative stress, and several studies have found a link between the consumption of specific foods or food groups and plasma/serum antioxidant capacity [83]. In women who have had breast cancer, it has been demonstrated that drinking fresh carrot juice on a daily basis is a simple and effective way to increase plasma total carotenoids and, as a result, reduce oxidative stress, but not inflammatory markers [84].

Additionally, in vitro evidence demonstrated the role of nuclear factor-kappa B (NF-κB), which has a critical role in cancer development and progression [85], as well as in regulating inflammatory pain [86]. Evidence also found that tomato extracts inhibited TNFα induced NF-κB activity in the androgen-independent human-derived prostate cancer cells [87].

Taking into account the links between chronic pain, inflammation, and metabolic dysregulation, and there subsequent impact in cancer survivors, a strategic dietary intervention for this population that could modulate this pathophysiology is worth looking into [47]. However, specific evidence for these mechanisms in cancer survivors is yet to be generated and represent an important area for future research.

### 3.4. Impact of Diet and Nutrition on Pain in Cancer Survivors through Gut Microbiota

In cancer cohorts, cancer treatments, specifically chemotherapy, has been proven to have a negative impact on the gut microbiome [88]. In support of this, the gut microbiota has been linked to psychoneurological symptoms associated with cancer treatment including chemotherapy-induced peripheral neuropathy by generating pro- and anti-inflammatory cytokines or chemokines to be produced [89]. Recent research suggests that dysbiosis of the gut microbiome is also a critical factor in central sensitization, which leads to chronic pain and cancer-related pain [90].

An increasing body of research demonstrates the critical function of gut microbiota in acute and chronic pain (neuropathic, inflammatory, and viscera) modulation and has ushered into a new era in pain management [91,92]. Additionally, gut microbiota-derived mediators in the central nervous system may modulate induction and maintenance of central sensitization via regulating neuroinflammation, which involves the activation of blood–brain barrier cells, microglia, and infiltrating immune cells [91,93,94]. Animal models have shown that gut microbes can stimulate the vagus nerve, which controls brain and behavior, and that changes in gut microbial composition are linked to significant changes in mood, pain, and cognition behaviors [95]. Preclinical and clinical findings suggest that communication between the gut microbiome, inflammation and microglia is involved in the development of chronic pain, implying that manipulating the gut microbiome in chronic pain sufferers could be an effective way to improve pain outcomes [96].

Dietary composition and amount play a significant role in gut microbiota composition and function [37]. It has been shown that lesser gut microbial diversity is associated with poorer nutritional status, frailty, comorbidity, and inflammation indicators [97]. Based on animal and human studies, dietary intake is seen to be a main short-term and long-term regulator of the gut microbiota structure and function [98]. To illustrate; some findings point to a relationship between Vitamin D insufficiency and altered nociception, presumably through molecular processes affecting the endocannabinoid and associated mediator signaling systems [99]. Additionally, short-chain fatty acids (SCFAs), which are microbial metabolites, interact with vagal afferents, and impact inflammation and hormonal control may also affect the peripheral immune system to modulate brain function [100]. Hence, targeting gut microbiota by dietary intervention is one of the innovative and possibly productive options for chronic pain therapy [91].

In cancer survivors, there are a few studies examining the relationship between nutrition and gut microbiota. For example, volunteers with a prior history of colorectal cancer who received rice bran or bean powder had increased gut bacterial diversity and altered gut microbial composition after 28 days when compared to baseline [101]. In overweight breast cancer survivors, probiotics in addition to a Mediterranean diet (MD) enhance gut microbiota and metabolic and anthropometric parameters as compared to an MD alone [37]. Another evidence of positive associations between the abundances of Bifidobacterium among the gut microbiota and the levels of omega-3 PUFAs in the blood came from a cross-sectional study that looked at the relationship between PUFAs and the gut microbiota among breast cancer survivors [102].

Understanding the link between the gut microbiota, nutrition and chronic pain has practical implications for cancer survivors. Nutrients initially meet the gut microbiota before being absorbed as bioactive products; hence anything related to the link between diet and pain is closely associated with the gut microbiome [24]. Guo et al. [91] assert that gut microbiota modulates pain in the peripheral and central nervous systems, and that targeting gut microbiota through diet and pharmabiotic intervention could be a new therapeutic approach for chronic pain treatment including chemotherapy-induced peripheral neuropathy pain.

## 4. How Can We Implement This Knowledge in Clinical Practice?

Studies in cancer survivors showed differences in nutrient intake status (such as total energy intake, carbohydrates and vitamin B) before and after cancer diagnosis [103] and in comparison to cancer survivors with non-cancer individuals [104]. Moreover, studies displayed poor adherence to diet recommendations, reports, and guidelines [105,106,107].

Latest literature recommends that nutritional counseling focused on micro-/macronutrients (such as vitamins, minerals, saturated fat, proteins, etc.) does not end up with adequate development in eating behaviors and may also cause an unnecessary increase in supplements consumption [108]. Additionally, due to reduced dietary intake and changed metabolism and absorption, cancer survivors often do not respond well to nutritional supplements [11]. For that reason, dietary supplement use is common and controversial among cancer survivors; however, evidence on the amount of nutrients received through supplements is lacking [109]. Still, cancer survivors reported a higher prevalence and dose of dietary supplement use, but lower nutritional consumption from foods, than those who had not been diagnosed with cancer [110]. However, there is some evidence that shows supplementation could work in cancer patients. For example, diet supplementation with some particular nutrients such as omega-3 fatty acids, vitamins B1, B3, B6, B12 and D, magnesium, zinc and β-carotene contributes to the alleviating of chronic pain as seen in systematic reviews in different pain populations including aromatase inhibitor-related arthralgia in breast cancer [55].

Since the cancer survivors population grows, clinicians as well as the healthcare system will require adapting and learning new methods to support patients with persistent pain [111]. Biologic therapies such as diet targeting the underlying causes of pain as discussed above could potentially reduce costs associated with cancer-related pain in survivors. Additionally, cancer survivors are often curious about food choices, physical activity and dietary supplements since they would like to learn whether nutrition and physical activity may help them live longer or feel better [112]. As part of this, nearly half of cancer survivors used nutritional supplements on their own without contacting their health care provider, which could indicate a lack of communication between cancer survivors and their health care providers about supplement use [110]. According to a study, only 2% of 1081 cancer survivors received dietary supplement counseling from a licensed dietitian [113]. In addition, promoting a dietary pattern rather than a specific food or nutrient may offer more health advantages, but future treatments must develop techniques that allow people to change numerous dietary habits successfully [73].

Furthermore, moderate-certainty evidence has demonstrated that dietary interventions can also modify food and nutrient intakes and positively affect some anthropometric measurements (such as weight loss and body mass index) in cancer survivors, especially in women after breast cancer [114]. In addition to that as discussed, some evidence found those positive changes could improve pain in (breast) cancer survivors [43,115]. In the light of these considerations, cancer survivors should also be encouraged to follow the recommendations for body weight as has been reported in a recent systematic review [116].

In a recent review that addressed adult cancer survivors’ perspectives on dietary advice following cancer treatment in the Australian context, cancer survivors reported a need for (a) individualized dietary strategies to address ongoing symptoms, (b) professional weight management support, and (c) practical skills for healthy eating [117]. Given that survivors are extremely motivated to improve their general health following a cancer diagnosis, healthy lifestyle recommendations from oncology providers can be a powerful motivator for survivors to embrace health behavior modifications [118]. A randomized controlled trial demonstrated that an education and culinary-based intervention in breast cancer survivors successfully increased adherence to a more anti-inflammatory dietary pattern by increasing consumption of anti-inflammatory foods, spices, and herbs while decreasing consumption of pro-inflammatory foods [73].

As a result, in order to make healthy lifestyle choices throughout survivorship, cancer survivors may benefit from additional advice and support [107]. However, cancer survivors receive a wide range of recommendations about what foods to consume or avoid, and what supplements to take, if any, from a variety of sources. Unfortunately, this advice is frequently inconsistent and unsupported by evidence [109]. However, even though more research is needed to completely integrate these approaches as management modality for chronic pain in cancer survivors, it is important to guide those people and inform them about the importance of healthy diet with so far accumulated evidence.

## 5. Conclusions

Obesity, malnutrition, nutritional deficiency, diet quality, immune system, systemic inflammation, and gut microbiota are some pathways/mechanisms associated with chronic pain in cancer survivors. As seen clearly, dietary interventions may provide weight reduction, a healthy body weight, good diet quality, regulations in systemic inflammation and immune system, and a healthy gut microbiota environment that could modify aforementioned pain-related pathways/mechanisms. For that reason, nutrition might have the potential to transition from being only prevention for cancer recurrence or cancer itself to a modality for chronic pain management for cancer survivors. In some available studies, nutrition has been already shown to improve survivors’ pain and quality of life (including bodily pain), which provides some basis and rationale considering the role of nutrition in chronic pain management carefully.

In the future, more clinical studies that directly explore nutrition and chronic pain in cancer survivors should be done to better understand the particular mechanisms that connect nutrition to chronic pain. Exploring and implementing awareness, prevention and management approaches that recognize the links between these elements is crucial to providing pain relief to survivors. 

## Figures and Tables

**Figure 1 jcm-11-00653-f001:**
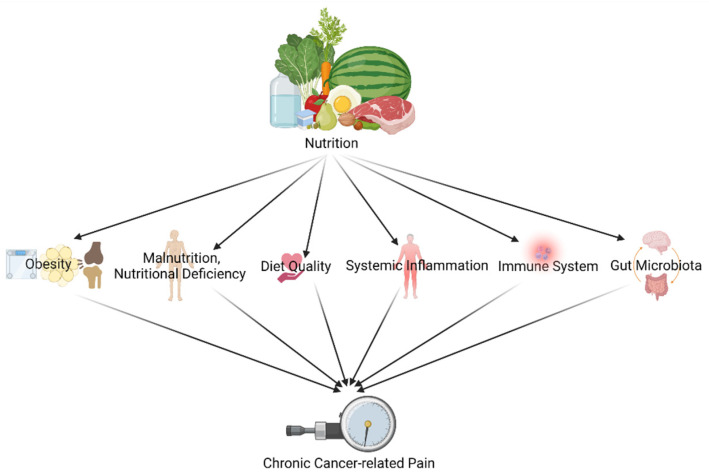
Different pathways and mechanisms to link diet/nutrition and (chronic) pain in cancer survivors (created with BioRender.com accessed on 23 December 2021).

**Table 1 jcm-11-00653-t001:** Contents of the dietary recommendations/guidelines for cancer survivors.

Reference	The Dietary Recommendations
2020 American Cancer Society (ACS) Guideline on Diet and Physical Activity for Cancer Prevention [66]	1. Achieve and stay at a healthy weight throughout life. Maintain a healthy body weight range throughout adulthood and avoid gaining weight. 2. Engage in physical activity. Adults should do 150–300 min of moderate-intensity physical activity each week, or 75–150 min of vigorous-intensity physical activity, or a combination of the two; reaching or beyond the upper limit of 300 min is ideal. Every day, children and adolescents should do at least 1 h of moderate- to vigorous-intensity activity. Limit sedentary behaviour such as sitting, lying down, and watching television or other screen-based entertainment. 3. Keep a healthy eating habit during your life. A healthy eating pattern comprises the following items: ○ Foods rich in nutrients in amounts that aid reach and maintain healthy body weight; ○ Various vegetables- dark green, red, and orange veggies, fibre-rich legumes (beans and peas), and others; ○ Fruit, particularly entire fruit in various colours; and ○ Whole grains. A healthy eating pattern excludes or restricts: ○ Meats, both red and processed; ○ Sugar-sweetened drinks; or ○ Refined grain products and highly processed foods. 4. It is better not to consume alcohol. Those who choose to consume alcohol should limit their intake to one drink per day for women and two drinks per day for males. Recommendation for Community Action At the national, state, and local levels, public, private, and community organizations should collaborate to develop, advocate for, and implement policy and environmental changes that increase access to affordable, nutritious foods; provide safe, enjoyable, and accessible opportunities for physical activity; and limit alcohol consumption for all people.
World Cancer Research Fund (WCRF)/American Institute for Cancer Research (AICR) Diet, Nutrition, Physical Activity and Cancer: a Global Perspective (2018) [67]	Maintain a healthy body weight Maintain a healthy weight and prevent gaining weight in adult life. Engage in physical activity Make physical activity a regular component of your everyday routine—walk more and sit less. Include whole grains, vegetables, fruit and beans in your diet Make a major part of your usual daily diet from whole grains, vegetables, fruit, and pulses (legumes) such as beans and lentils. Limit intake of ’fast foods’; and other processed foods that are high in fat, starches or sugars Limiting these foods can help you keep track of your calorie intake and maintain a healthy weight. Limit red and processed meat consumption. Red meat, such as beef, pork, and lamb, should be consumed in moderation. Consume very little, if any, processed meat. Limit sugar-sweetened drinks consumption Drink usually water and non-sweetened drinks. Limit consumption of alcohol. It’s best to not consume alcohol for preventing cancer. Supplements should not be used to prevent cancer Aim to achieve nutritional needs solely through diet. If you are a mother: if you are able, breastfeed your baby. Breastfeeding is beneficial to both mother and baby. After recieving a cancer diagnosis: if you are able, follow our recommendations. Consult your medical providers to determine what is right for you.

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
