# Peer review of "Diet/Nutrition: Ready to Transition from a Cancer Recurrence/Prevention Strategy to a Chronic Pain Management Modality for Cancer Survivors?"

_jcm, 2022, doi:10.3390/jcm11030653_

Round 1

Reviewer 1 Report

First of all, I would like to thank and congratulate the authors for this work. Although it is a topic poorly addressed in the scientific literature, the authors have managed to synthesize valuable information that will undoubtedly contribute to further progress in this interesting field. Below are some suggestions that in my opinion can improve the article, although the authors are free in their own discretion to adopt it:

  1. I think that the abstract is excessively general. It would be interesting to comment briefly on some of the links or mechanisms proposed between nutrition and chronic pain management in cancer survivors.
  2. The introduction could elaborate more on the common relationships and mechanisms between chronic pain, cancer and nutrition. It is not entirely clear why these patients are different from other types of chronic pain, and why nutritional strategies may play a role in their management (and less so with a recent systematic review finding no evidence).
  3. Related to the previous point, I think it might be interesting to comment on some mechanisms that could be related in chronic pain and cancer, such as oxidative stress or mitochondrial dysfunction. I would recommend going deeper along the lines of some articles that the authors are familiar with such as: DOI: 10.3390/ijms16010193 DOI:10.3390/antiox9111166 DOI:10.1517/14728222.2013.818657 DOI: 10.1136/postgradmedj-2012-131068
  4. When talking about diet quality, I think it might be interesting to talk about nuclear factor-kappaB, given that previous studies have found relationships between this factor and chronic inflammation in patients with cancer or chronic pain. DOI: https://doi.org/10.1016/j.pbb.2015.04.005 DOI: 10.1016/j.bbi.2015.07.014 DOI: 10.1080/01635581.2015.990575 DOI: 10.1016/j.nutres.2013.07.008

Author Response

Response to Reviewer 1 Comments

Point 1: I think that the abstract is excessively general. It would be interesting to comment briefly on some of the links or mechanisms proposed between nutrition and chronic pain management in cancer survivors.

Response 1: Thank you for the supportive comments and suggestion. The requested links/ mechanisms have been added into abstract.

This now reads: “Given the relationship between chronic pain, inflammation, and metabolism found in the literature, it is advised to look for a strategic dietary intervention in cancer survivors. Dietary interventions may result in weight loss, a healthy body weight, good diet quality, systemic inflammation and immune system regulations, and a healthy gut microbiota environment, all of which may alter the pain-related pathways and mechanisms.”

Point 2: The introduction could elaborate more on the common relationships and mechanisms between chronic pain, cancer and nutrition. It is not entirely clear why these patients are different from other types of chronic pain, and why nutritional strategies may play a role in their management (and less so with a recent systematic review finding no evidence).

Response 2: Thank you for the suggestion. Because the next sections of the paper include the requested details and information, we kept the introduction section brief to avoid repetitions. However, to account for your request, we added more general information to the introduction.

This section now reads: “Persistent pain in cancer survivors is often complex (neuropathic, nociplastic, and/or nociceptive) in nature [1], underrecognized, undertreated and less responsive to regular chronic pain management approaches (i.e., pharmacological treatments, rehabilitation, etc.) [2]. Additionally, in long-term survivors (in comparison to people without a history of cancer), it is known that the incidence and relative risk of chronic comorbidities is high [3], which result in significantly more functional limitations and pain intensity, making them less likely to respond to standard chronic pain treatment [4].

                                         AND

“Nutritional medicine for treating persistent pain requires a comprehension of the disease process’ pathogenesis that helps practitioners to prescribe ingredients with particular roles in alleviating the disease process, such as inflammation reduction, or with particular influences on other factors which contribute to pain (i.e., stress and insomnia) [5], or oxidative stress-modulating compounds and oxidative stress status [6].”

Point 3: Related to the previous point, I think it might be interesting to comment on some mechanisms that could be related in chronic pain and cancer, such as oxidative stress or mitochondrial dysfunction. I would recommend going deeper along the lines of some articles that the authors are familiar with such as: DOI: 10.3390/ijms16010193 DOI:10.3390/antiox9111166 DOI:10.1517/14728222.2013.818657 DOI: 10.1136/postgradmedj-2012-131068

Response 3: Thank you for the suggestion. We’ve added the requested information to the content of the paper: “Furthermore, mitochondrial dysfunction caused by cancer cells (induced by the mitochondrial genome alterations, the associated oxidative stress etc.) [7] may play a role in chronic pain. Maintaining mitochondrial function has been proposed as a possible treatment technique for treating or preventing chronic pain [8]. For example, strategies that improve mitochondrial function have shown success in preventing and reversing CIPN in pre-clinical animal models and have begun to show some progress toward translation to the clinic [9].”

Point 4: When talking about diet quality, I think it might be interesting to talk about nuclear factor-kappaB, given that previous studies have found relationships between this factor and chronic inflammation in patients with cancer or chronic pain. DOI: https://doi.org/10.1016/j.pbb.2015.04.005 DOI: 10.1016/j.bbi.2015.07.014 DOI: 10.1080/01635581.2015.990575 DOI: 10.1016/j.nutres.2013.07.008

Response 4: Thank you for the suggestion. We have added the requested information by adding the following text: “Additionally, in vitro evidence demonstrated the role of nuclear factor-kappa B (NF-κB), which has a critical role in cancer development and progression [10], as well as in regulating inflammatory pain [11]. Evidence also found that tomato extracts inhibited TNFα induced NF-κB activity in the androgen-independent human-derived prostate cancer cells [12].”

  1. Leysen L, Adriaenssens N, Nijs J, Pas R, Bilterys T, Vermeir S, et al. Chronic Pain in Breast Cancer Survivors: Nociceptive, Neuropathic, or Central Sensitization Pain? Pain Practice. 2018.
  2. Burton AW, Fanciullo GJ, Beasley RD, Fisch MJ. Chronic pain in the cancer survivor: a new frontier. Pain Med. 2007;8(2):189-98.
  3. Chao C, Bhatia S, Xu L, Cannavale KL, Wong FL, Huang P-YS, et al. Chronic comorbidities among survivors of adolescent and young adult cancer. Journal of Clinical Oncology. 2020;38(27):3161.
  4. Paice JA, Portenoy R, Lacchetti C, Campbell T, Cheville A, Citron M, et al. Management of chronic pain in survivors of adult cancers: American Society of Clinical Oncology Clinical Practice Guideline. Journal of Clinical Oncology. 2016;34(27):3325-45.
  5. Audette JF, Bailey A. Integrative pain medicine: the science and practice of complementary and alternative medicine in pain management: Springer Science & Business Media; 2008.
  6. Hendrix J, Nijs J, Ickmans K, Godderis L, Ghosh M, Polli A. The interplay between oxidative stress, exercise, and pain in health and disease: potential role of autonomic regulation and epigenetic mechanisms. Antioxidants. 2020;9(11):1166.
  7. Luo Y, Ma J, Lu W. The significance of mitochondrial dysfunction in cancer. International journal of molecular sciences. 2020;21(16):5598.
  8. Sui B-d, Xu T-q, Liu J-w, Wei W, Zheng C-x, Guo B-l, et al. Understanding the role of mitochondria in the pathogenesis of chronic pain. Postgraduate medical journal. 2013;89(1058):709-14.
  9. Doyle TM, Salvemini D. Mini-Review: Mitochondrial dysfunction and chemotherapy-induced neuropathic pain. Neuroscience letters. 2021;760:136087.
  10. Schatz AA, Oliver TK, Swarm RA, Paice JA, Darbari DS, Dowell D, et al. Bridging the Gap Among Clinical Practice Guidelines for Pain Management in Cancer and Sickle Cell Disease. Journal of the National Comprehensive Cancer Network. 2020;18(4):392-9.
  11. Hartung JE, Eskew O, Wong T, Tchivileva IE, Oladosu FA, O’Buckley SC, et al. Nuclear factor-kappa B regulates pain and COMT expression in a rodent model of inflammation. Brain, behavior, and immunity. 2015;50:196-202.
  12. Kolberg M, Pedersen S, Bastani NE, Carlsen H, Blomhoff R, Paur I. Tomato paste alters NF-κB and cancer-related mRNA expression in prostate cancer cells, xenografts, and xenograft microenvironment. Nutrition and cancer. 2015;67(2):305-15.

Reviewer 2 Report

  1. Very generic review
  2. No strong data or good evidence provided
  3. Oversimplification or General good pricinples used to overempahsize the role of diet in chronic pain amanegment in cancer patient. 
  4. would be helpful if any study is icnluded that shows diet management resulted in a better chemoinduced pain control.

Author Response

Response to Reviewer 2 Comments

Point 1: Very generic review

Response 1: Thank you for the comment. Unfortunately, given the limited evidence available, we were indeed not able to provide more details or go deeper in the light of the complexity involved in diet/ nutrition, chronic pain mechanisms, and cancer-related factors (type, stage, treatment modalities etc.).

Point 2: No strong data or good evidence provided

Response 2: Thank you for the comment. We fully agree with your notion. The reviewer points out a fact, which is one of the main reasons for conducting this study and one of the main messages of the paper.

Point 3: Oversimplification or General good principles used to overemphasize the role of diet in chronic pain management in cancer patient.

Response 3: Thank you for the positive feedback. Since there are various possible mechanisms to be emphasized and considered to point out the possibility of using diet/ nutrition in chronic pain management in cancer survivors, the sections could be seen as repetitive or overemphasised. The paper has been structured like this to attract the attention of clinicians and researchers on the topic.

Point 4: would be helpful if any study is included that shows diet management resulted in a better chemo induced pain control.

Response 4: Thank you for the suggestion. We agree that including studies that have findings regarding chemo-induced pain management by using diet/ nutrition as the modality in cancer patients/ survivors could help enriching our discussion. Unfortunately, according to our research currently, no studies are available on this specific topic in this particular population. However, to further address this important issue, we have added the following sentence to the paper: “Besides that, studies examining whether dietary management results in pain relief in cancer patients receiving chemotherapy or in survivors after treatment (or in any other cancer treatment associated with pain) are urgently needed and represent an important research priority.”.
